# Real-World Analysis of Durvalumab after Chemoradiation in Stage III Non-Small-Cell Lung Cancer

**Beatrice T. B. Preti** [1,2] 📷, **Michael S. Sanatani** [1,2], **Daniel Breadner** [1,2,*] 📷, **Suganija Lakkunarajah** [3], **Carolyn Scott** [1], **Caroline Esmonde-White** [1], **Eric McArthur** [4], **George Rodrigues** [1,5], **Mitali Chaudhary** [6], **Adam Mutsaers** [1,5], **Robin Sachdeva** [3] and **Mark D. Vincent** [1,2]

1 Schulich School of Medicine & Dentistry, Western University, London, ON N6A 3K7, Canada; bpreti@qmed.ca (B.T.B.P.)
2 Division of Medical Oncology, Schulich School of Medicine & Dentistry, Western University, London, ON N6A 3K7, Canada
3 Department of Medical Oncology, University of British Columbia, Victoria, ON V8R 6V5, Canada
4 London Health Sciences Centre, London, ON M5S 1A8, Canada
5 Division of Radiation Oncology, Schulich School of Medicine & Dentistry, Western University, London, ON N6A 3K7, Canada
6 Temerty School of Medicine, University of Toronto, Toronto, ON M5S 1A1, Canada
* Correspondence: daniel.breadner@lhsc.on.ca

**Abstract:** The 2017 PACIFIC trial heralded the incorporation of routine adjuvant durvalumab following curative-intent chemoradiation for stage III non-small-cell lung cancer (NSCLC). However, carefully selected clinical trial populations can differ significantly from real-world populations, which can have implications on treatment toxicities and outcomes, making it difficult to accurately counsel patients. Consequently, we performed a real-world, retrospective analysis of outcomes and toxicities in 118 patients with stage III NSCLC treated with durvalumab after platinum-based chemoradiotherapy. The data were collected from patients who underwent treatment at a single, tertiary-level Canadian cancer centre from May 2018 to October 2020. The variables collected included patient demographics, treatment specifics, progression-free survival, overall survival, and immune-related adverse events (IRAE) from durvalumab. Descriptive statistics were used for toxicity analysis, and progression-free survival and overall survival estimates were calculated using the Kaplan–Meier method. The statistical analyses indicated a 64.4% ($n = 76$) toxicity rate, with a 21% ($n = 25$) toxicity rate of grade 3+ IRAEs. The most common documented IRAEs were pneumonitis ($n = 44$; 40%), followed by rash ($n = 20$; 18%) and thyroid dysfunction ($n = 17$; 15%). FEV1 and DLCO were not found to be associated predictors of pneumonitis toxicity. The median PFS and OS were estimated to be >1.7 years and >2.7 years, respectively.

**Keywords:** durvalumab; non-small-cell lung cancer; stage III lung cancer; immunotherapy; pneumonitis

## 1. Introduction

Stage III non-small-cell lung cancer (NSCLC) represents a heterogeneous class of disease which continues to serve the oncology community a treatment quandary; often, such tumours are not resectable, yet the specifics of optimal multi-modality treatment remain elusive [1]. In the pre-immunotherapy era, the foundation of managing unresectable stage 3 NSCLC was sequencing platinum-doublet chemotherapy regimen with radiation, where concurrent chemoradiation is superior to sequential treatment. The 2017 PACIFIC trial heralded the incorporation of consolidative durvalumab, an intravenous programmed cell death 1 receptor (PD-1) inhibitor, following curative-intent chemoradiation for stage III NSCLC [2]. This trial successfully demonstrated statistically and clinically significant increased median progression-free survival (PFS) (16.8 months vs. 5.6 months, hazard ratio 0.55) and median time to death or distant metastases (23.2 months vs. 14.6 months,

hazard ratio 0.72) in patients with stage III NSCLC who received up to 12 months of consolidative durvalumab vs. placebo after platinum-based chemotherapy in the context of concurrent chemoradiation [2]. In the 5-year analysis, it did show a durable survival benefit with the median overall survival (OS) being 47.5 months vs. 29.1 months, favouring the durvalumab group [3]. Adverse effects of any grade were documented in 96.8% of the durvalumab group and 94.9% of the control group, with cough (35.4% vs. 25.2%) and all-cause pneumonitis (33.9% vs. 24.8%) being the most frequently reported adverse effects. Absolute increased adverse effect risk was quoted at 1.9% higher in the durvalumab group, whereas grade 3–4 immune-related adverse effects (IRAEs) were 3.8% higher in the durvalumab group, a manageable risk–benefit when considering immunotherapy.

However, carefully selected clinical trial populations can differ significantly from populations in routine clinical practice, which can have implications on treatment toxicities and outcomes [4]. Real-world populations and outcomes can also differ depending on the specific population(s) studied, and studying real-world populations has its own intrinsic value in cancer care, helping both clinicians and patients weigh the costs and benefits of treatments.

Since the publication and implementation of the PACIFIC trial protocol, several real-world studies have been performed across the globe. For instance, Sankar and colleagues demonstrated similar PFS in a real-world population, while the OS benefit was lower, likely related to differences in the patient population compared to the PACIFIC study [5]. As seen in the PACIFIC study, similar real-world studies showed moderate rates of pneumonitis, a significant cause of discontinuation due to IRAEs. There have been some studies that showed some association with smoking or presence of COPD as risk factors for developing pneumonitis on immunotherapy agents, but very little is known about whether pretreatment baseline lung function tests are valuable as risk or prognostic factors associated with the development of pneumonitis [6].

In this single-centre, retrospective study, we reviewed the safety and survival outcomes of consolidative durvalumab after platinum-based chemoradiation. We focused on key patient factors to assess any potential risk factors for toxicity, in particular, immune-related pneumonitis.

## 2. Methods

### 2.1. Objective(s)

Primary objectives:

(1)   To compare the PACIFIC trial toxicity outcomes to the outcomes in a real-world, non-trial-based population.
(2)   To determine whether the efficacy outcomes of the PACIFIC trial are maintained in a real-world analysis.

Secondary objectives:

(1)   To determine whether there are risk factors for (severe) IRAEs from durvalumab.

### 2.2. Study Details

This retrospective study was reviewed and approved by Western University's research ethics board (REB, ReDA 118353). Data were collected from patients with stage III NSCLC who underwent treatment with consolidative durvalumab following curative-intent concurrent chemoradiation at a single tertiary-level Canadian cancer centre, with Health Canada approval in 2018 until October 2020; this included the provincial funding approval received in March 2020. Due to ethics constraints, events were captured up to June 2021. A total of 136 patient charts were identified through medical records for having been scheduled for durvalumab; of these, 118 patients were identified as fitting our criteria (below) and included in the analysis. The variables collected included patient demographics, treatment specifics, progression-free survival, overall survival, and toxicities from durvalumab. A

complete list of variables collected is included in Appendix A. PD-L1 expression was calculated using the Dako 22C3 PD-L1 IHC PharmDx assay.

IRAEs were graded using the Common Terminology Criteria for Adverse Events (CTCAE v5) based on the clinical notation available in the patient charts. In this study, immune-related pneumonitis was defined as an event of pneumonitis occurring after at least 1 cycle of durvalumab, in an effort to try to avoid collecting data from patients with (purely) radiation-driven pneumonitis.

The patient inclusion and exclusion criteria are detailed below.

Patient criteria:

(1)   Patient inclusion criteria:

- Histologic/cytologic diagnosis of stage III non-small-cell lung cancer.
- Received two or more cycles of platinum-based chemotherapy, at least one of which was given concurrently with radiation.
- No disease progression observed during chemoradiation or between chemoradiation completion and start of immunotherapy.
- Received at least one dose of durvalumab.
- Treated locally at our tertiary-level centre.

(2)   Patient exclusion criteria:

- Identification of metastatic disease prior to start of treatment.

*2.3. Analysis*

Analysis was performed mainly using descriptive statistics. IRAE outcomes were assessed as binary outcomes with proportions presented, as well as their associated 95% confidence intervals. Efficacy outcomes (PFS and overall survival [OS]) were assessed using survival curves estimated using the Kaplan–Meier product limit estimator. Cumulative probability of the outcomes at various time points and median survival were also estimated based on the Kaplan–Meier curves.

Based on these findings, we calculated the odds ratios to assess potential risk factors for toxicities.

**3. Results**

From 2018 to 2020, 118 patients with a diagnosis of locally advanced non-small-cell lung cancer were treated with concurrent chemoradiation followed by consolidative durvalumab. The majority of these patients had an ECOG status of 0 or 1, as listed in Table 1. The male/female ratio was roughly equal, unlike many other studies, including the PACIFIC trial itself, which had a 70% male predominance. Of note, our rates of smoking were significantly higher than the PACIFIC trial (current/former/never 36%/48%/4% vs. 16%/75%/9%). The performance status scores were also notably lower (ECOG 0/1/2/3 27%/34%/5%/3% vs. 49%/51%/0%/0%). The majority of the patients had either stage IIIA or IIIB disease; some stage IV patients at our centre underwent treatment with curative-intent local therapy with chemoradiation followed by durvalumab, and these were included in our analysis. The patients with stage IV disease underwent SBRT for oligometastatic lesions.

The median time from end of chemoradiation (defined as last day of radiation) to durvalumab start was 40.5 days. In this patient population, the shortest amount of time from chemoradiation to durvalumab start was 7 days and the longest was 238 days. In total, 53 patients (44.9%) began treatment more than 42 days after the completion of chemoradiation. The median follow-up was 534 days (just under 1.5 years).

At the time of analysis, 86 patients out of 118 were alive. The statuses of six patients were unknown as they were lost to follow-up. Of the 26 patients who died, 14 patients died due to disease recurrence. Death related to non-oncologic/toxicity causes was not recorded.

**Table 1.** Demographic Information.

| Age (Mean in Years) | 66.3 |
|---|---|
| Sex | |
|     Male | 61 |
|     Female | 57 |
| **ECOG** | |
|     0 | 32 |
|     1 | 40 |
|     2 | 6 |
|     3 | 3 |
|     Unknown/Not Done | 37 |
| **Histology Type** | |
|     Adenocarcinoma | 73 |
|     Squamous Cell Carcinoma | 38 |
|     Other | 7 |
| **Chemo Received** | |
|     Carboplatin/Paclitaxel | 40 |
|     Cisplatin/Etoposide | 32 |
|     Carboplatin/Vinblastine | 19 |
|     Cisplatin/Vinblastine | 10 |
|     Cisplatin/Vinorelbine | 1 |
|     Other | 16 |
| **Status** | |
|     Dead | 26 |
|     Alive | 86 |
|     Unknown | 6 |
| **Smoking Status** | |
|     Current | 41 |
|     Former | 66 |
|     Never | 5 |
|     Unknown | 3 |
| **Stage** | |
|     IIIA | 47 |
|     IIIB | 49 |
|     IIIC | 6 |
|     IV | 7 |
|     Other/Unknown (including stage II) | 6 |
| **Disease Recurrence?** | |
|     Yes | 45 |
|     No | 73 |
| **PDL1 Status** | |
|     <1% | 29 |
|     1–49% | 38 |
|     ≥50% | 36 |
|     Unknown | 15 |
| **Time from CRT completion to durvalumab start (median time)** | 40.5 days |

Due to a lack of events, we were unable to calculate specific median progression-free survivals and overall survivals, but these were estimated to be >20 months and >32 months, respectively. The Kaplan–Meier curves, both overall and stratified by PD-L1 status, are shown in Figure 1. There were no statistical differences between the PD-L1-positive and PD-L1-negative patients.

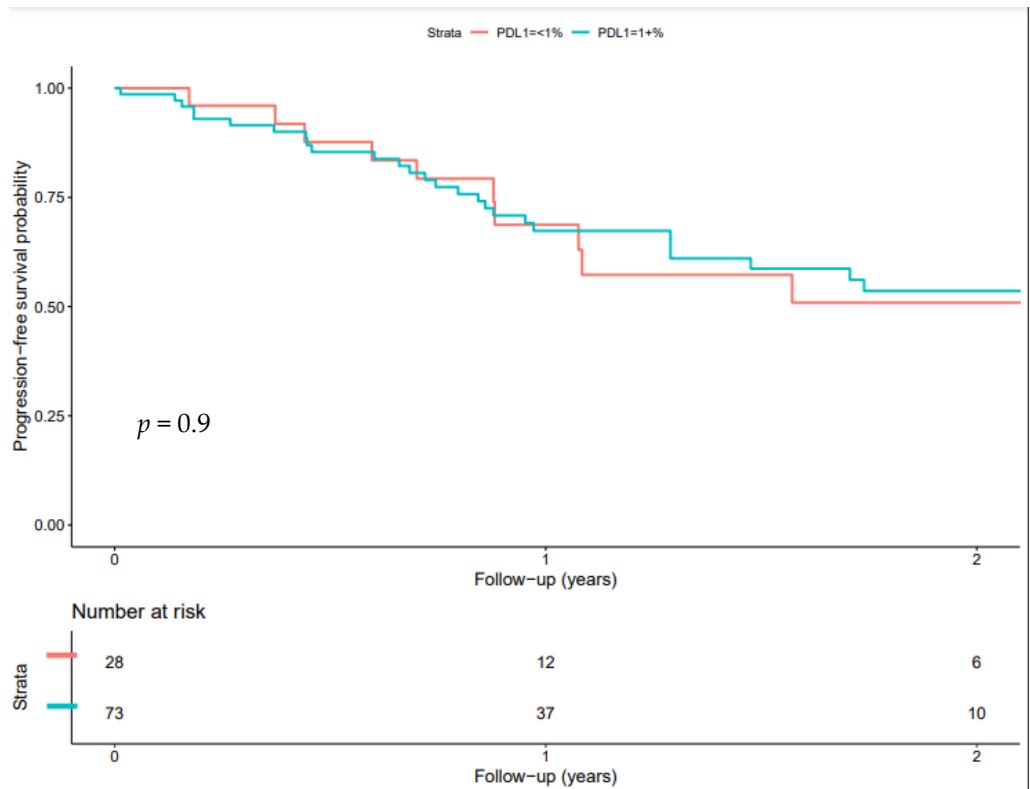

**Figure 1.** Progression-free survival.

### 3.1. Immune-Related Adverse Events (IRAEs)

In total, 64.4% (*n* = 76) of patients developed at least one IRAE of any grade, with 21% (*n* = 25) developing grade 3+ toxicities. Three patients (2.5%) died due to toxicity while on treatment, all from immune-related pneumonitis. A total of 50 (42.4%) patients stopped treatment prematurely due to severe IRAEs. Another 25 patients (21.2%) experienced a treatment delay due to an event but were subsequently rechallenged on durvalumab. The most common documented IRAEs occurring during treatment were pneumonitis (*n* = 44 events, 40%), followed by rash and thyroid dysfunction (*n* = 20 and 17 events, 18.2% and 15.5%, respectively). A full list of toxicities and frequencies can be found in Table 2.

**Table 2.** Toxicities.

| Toxicity Events, by Type | Any Grade | Grade I | Grade II | Grade III | Grade IV | Grade V |
|---|---|---|---|---|---|---|
| Skin rash | 20 | 9 | 9 | 2 | 0 | 0 |
| Thyroid dysfunction | 17 | 4 | 13 | 0 | 0 | 0 |
| Pleuritis | 0 | 0 | 0 | 0 | 0 | 0 |
| Pneumonitis | 47 | 4 | 26 | 12 | 5 | 3 |
| Heart inflammation | 2 | 0 | 1 | 1 | 0 | 0 |
| Colitis proctitis | 10 | 3 | 6 | 1 | 0 | 0 |
| Pancreatitis | 0 | 0 | 0 | 0 | 0 | 0 |
| Hypophysitis | 0 | 0 | 0 | 0 | 0 | 0 |
| Hepatitis | 1 | 1 | 0 | 0 | 0 | 0 |
| Arthritis | 1 | 0 | 0 | 1 | 0 | 0 |
| Other | 12 | 3 | 3 | 5 | 1 | 0 |
| **Total** | 110 | 24 | 58 | 22 | 6 | 3 |

We noted our 40% pneumonitis rate (14.4% for grade 3/4) is numerically higher from the quoted 33.9% for all-grade and 3.4% for grade 3/4 rate of pneumonitis in the PACIFIC trial study. In an attempt to exclude clear radiation-induced pneumonitis as much as we were able to, although it should be understood that the delayed-onset nature of radiation pneumonitis can make it difficult to distinguish from durvalumab-induced pneumonitis, we only included cases of pneumonitis diagnosed after at least one dose of durvalumab.

*3.2. Risk Factors*

Based on our findings, we were specifically interested in any potential risk factors for pneumonitis, as this appeared to be not only the most common toxicity, but also the most common grade 3+ toxicity and cause of all IRAE-related deaths. Consequently, we calculated the odds ratios for the association between pneumonitis and FEV1 and DLCO for 61 patients with available pulmonary function tests (PFTs). The odds ratio for FEV1 or DLCO was 1.00 (95% CI 0.98–1.03, $p = 0.91$) and 1.00 (95% CI 0.97–1.03, $p = 0.95$), respectively, which suggests against any potential relationship. Other factors, such as total radiation dose, total radiation fractions, radiation planning treatment volume (PTV), radiation gross tumour volume, V30, V20, V5, and mean lung dose, were initially going to be analyzed; but due to the poor documentation of this information, there were not enough data points to look for any possible association with the risk of developing pneumonitis.

**4. Discussion**

We present the data for a single-centre, retrospective real-world analysis examining the use of durvalumab after chemoradiation in stage III NSCLC, including statistical analyses examining baseline lung function as a risk factor for pneumonitis.

The PACIFIC trial heralded the use of consolidative durvalumab for patients with stage III NSCLC who were treated with curative-intent chemoradiation. This led to a remarkable improvement in the survival outcomes for patients, both in terms of PFS and OS.

In this current series, the patient population studied had a few similarities to the original PACIFIC study population and other real-world analyses. The average age was in the mid-60s, with most patients having an ECOG status of 0–1. Regarding the ECOG status, this might be due to a selection bias as those with a better performance status or deemed healthier would be offered concurrent treatment, while those with a poorer status would be offered sequential or palliative treatment and then not have access to durvalumab and not be included in these studies.

Multiple studies had similar population distribution regarding age and performance status [7]. Some differences exist between this patient population and those in previous studies with regard to gender and smoking status. The population studied in this paper included a relatively equal number of male and female patients compared to other larger retrospective studies and the PACIFIC trial. This might help explain the decreases in survival or increase in toxicities as previous systematic reviews have shown that these outcomes can be gender-related [8].

In addition, a higher proportion of our patient population was either current or former smokers in comparison to similar study populations.

The efficacy outcomes seen in our study do not appear to be inferior to the PACIFIC study outcomes, an important finding for real-world evidence, although we were only able to estimate the values, which leaves room for uncertainty. That said, our findings show improved outcomes compared to historical data, but less benefit than what other real-world analyses have found [4,9]. In the PACIFIC-R study, the median rwPFS was 21.7 mo while the mOS was still not met, with an estimation of 70% of the population being alive in 2 years. The slightly higher median rwPFS compared to the results of the PACIFIC trial is likely related to the less strict radiographic follow-up of patients while on therapy in the community [9]. Conversely, at approximately 5-year follow-up, the mOS in the study by Sankar et al. was numerically lower than the PACIFIC study (34.7 mo vs. 47.5), which is likely explained by the slightly older patient population with increased comorbidities [5].

Our study found equivalent OS outcomes for patients regardless of the PD-L1 status. This might suggest that real-world patients without PD-L1 tumour expression are different compared to trial patients without PD-L1 tumour expression, which is supported by other literature showing minimum differences between the two groups [10]. We do note that a less-rigorous imaging follow-up schedule compared to previous clinical trials might have artificially increased our reported PFS; however, we do see improved OS compared to historical data, which would not be subjected to this bias.

This study showed a higher rate of all-grade and grade 3+ immune-related pneumonitis compared to the PACIFIC trial. These results are supported by other real-world analyses [11]. All three on-treatment deaths were from pneumonitis. We also observed higher rates of all-grade and grade 3+ rash and thyroid dysfunction. These were the three most common durvalumab toxicities noted by our study. As mentioned earlier, there could be several factors contributing to this, including a high proportion of current (36%) and former (57%) smokers, as well as less fit patients with less pulmonary reserve. However, a subsequent statistical analysis revealed that FEV1 and DLCO were not predictors of pneumonitis toxicity.

What is interesting to note is that most studies do not characterize pneumonitis events as either being radiation- or immune-related events as it difficult to distinguish between the two. In this series, immune-related pneumonitis was defined temporally based on the onset after the first dose of consolidative durvalumab. Radiation pneumonitis can occur up to 4 to 12 weeks after radiation exposure [12]. The overly simplistic definition used in this study might have captured non-immune-related pneumonitis, which might account for the higher incidence of events compared to the PACIFIC trial.

The limitations of our study include the single-centre nature and relatively limited number of patients being treated. Other limitations related to this study being a retrospective study are the variability of documentation and missing information (ECOG status, etc.). We did collect information such as treatment after progression/cessation of consolidative durvalumab as this might impact overall survival rate or EGFR status [13]. We also included toxicities as diagnosed, documented, and graded by the treating oncologist in an attempt to standardize patient records, as not every patient received subspeciality input or further testing for all toxicities.

Potential next steps could include further analyses investigating further risk factors or potential indicators for pneumonitis. Additionally, new evidence for neoadjuvant chemoimmunotherapy might pose a new focus of real-world study in the lung cancer/immunotherapy population [14,15].

## 5. Conclusions

In summary, this analysis confirmed the survival benefits of the PACIFIC trial in a real-world setting, although it did show high rates of pneumonitis. We were not able to identify risk factors for pneumonitis; specifically, differences in baseline lung function tests were not predictive for the development of immune-related pneumonitis.

**Author Contributions:** D.B., M.D.V., M.S.S. and G.R. were involved in project conception and design. B.T.B.P., S.L., C.S., M.C., C.E.-W. and A.M. were involved in data collection and entry. R.S. and E.M. were involved in data analysis. The manuscript was written by D.B., S.L. and B.T.B.P., with input from all listed authors. All authors have read and agreed to the published version of the manuscript.

**Funding:** This research received no external funding.

**Conflicts of Interest:** The authors declare no conflict of interest.

**Appendix A**

The list of variables which were collected via retrospective chart review are described below:

- Patient demographics: age at diagnosis, date of birth (month/year only), sex, disease stage (TNM), Eastern Cooperative Oncology Group—Performance Status score, tumour histologic type, smoking status (current/former/never), total radiation dose, total radiation fractions, radiation planning treatment volume (PTV), radiation gross tumour volume, V30, V20, V5, mean lung dose, FEV1, FVC, DLCO, DLCO/VA, tumour lobe location, delays in radiation, actual dose and fractions delivered vs. planned, radiation adverse effects, prescribed creams for skin reaction, PPI/Viscous lidocaine/Sulcrafate prescription during treatment for esophagitis, time from radiation completion to first dose of durvalumab, chemotherapy regimen received, delays in chemotherapy, chemotherapy adverse effects necessitating medical action (delay/cessation/prescription medication), worst chemotherapy adverse effect, response to concurrent chemoradiation (complete/partial/stable disease), toxicities from chemotherapy persisting into the start of durvalumab treatment, comorbidities (COPD, ILD, autoimmune disorders, arthritis), and PDL1 expression.
- Outcomes measured from the start of chemotherapy and adjuvant durvalumab: progression-free survival, overall survival, and duration on durvalumab.
- Adverse effects: Presence, grade, and timing of symptoms (cough, fatigue, dyspnea, diarrhea, altered appetite, nausea, arthralgias, pruritis, constipation, headache, asthenia, MSK pain) and conditions (pneumonia, pneumonitis, pyrexia, arthritis, rash, hypothyroidism, anemia, and other immune toxicity). Grading based on CTC-AE version 5.0, and actionable vs. non-actionable toxicities. Actionable toxicities were further delineated into requiring treatment delay, non-PO/IV corticosteroid intervention, PO/IV corticosteroid intervention, treatment cessation, hospitalization, ICU-level care, or death.

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
