# Peer review of "Real-World Analysis of Durvalumab after Chemoradiation in Stage III Non-Small-Cell Lung Cancer"

_curroncol, doi:10.3390/curroncol30080559_

Round 1
Reviewer 1 Report
See the attached document.

Reviewer 2 Report
The manuscript entitled “Real-world analysis of durvalumab after chemoradiation in stage III non-small cell lung cancer” describes a single-center, retrospective study which investigated the toxicity and efficacy of durvalumab consolidation after chemoradiotherapy in a real-world population.
The study itself is well organized. However, there are several retrospective studies on this issue.
As in Discussion part, the PACIFIC-R study has elucidated lots of real-world evidence, including pneumonitis. And the real-world evidence focusing on pneumonitis has already been reported (Lung Cancer 2021; 161: 86-93.).
Therefore, the authors should show the originality in the current study other than the risk for pneumonitis.
I suggest the authors for major revision based on this point.
Reviewer 3 Report
This is a manuscript analyzing real-word data of the use of durvalumab, a PD-1 inhibitor, against Stage 3 Non-Small Cell lung cancer (NSCLC). Main objectives were trial toxicity profile of durvalumab and efficacy after initial standard of care chemoradiotherapy. This is a nice study with important findings.
There are a few remarks/comments to empower the demonstrated results:
- In the immune-related AEs section a comment on the proportion of subjects still smoking could be added to elucidate results regarding pneumonitis.
- In Figure 1 the survival curves demonstrate patients on durvalumab with PDL<=1% AND PDL>1%. Please clarify indication for PD-1 inhibitors when PDL<=1%.
- Did any patient receive any other EFGR or ALK targeting drug? Please clarify in the methods section.
- Please explain in the methods section how pneumonitis diagnosis was established. Since your study is retrospective this clarification is essential, because GGOs misinterpreted as pneumonitis in historical chest CTs could influence results.
Reviewer 4 Report
The scope of this paper is to assess the real world experience with consolidation durvalumab given in context of concomitant chemo/radiotherapy for NSCLC. The paper is relevant because the efficacy and safety of the addition of durvalumab was established in the Pacific trial which like most trial were conducted at large volume and/or academic sites where the standards of care and patient population may differ from more average populations seen generally. The study is retrospective and includes 118 patients.
Interestingly, but perhaps to be expected, the RWE group had a larger smoking exposure and a lower PS
Material: Since this is a comparison with PACIFIC it could be useful to expand table 1 to include PACIFIC numbers. Same for table 2 though perhaps lumping Grade groups into 1-2 and 3+ to make the table easier to read.
Discussion is good and I agree to pneumonitis to of particular interest
Reviewer 5 Report
Clinical trial populations are generally different from populations in the real world. The outhors performed a real-world, retrospective analysis of outcomes and toxicities in 118 patients with stage III NSCLC treated with durvalumab after platinum-based chemoradiotherapy. The analysis confirms the survival benefits of the PACIFIC trial in a real world setting though showed high rates of pneumonitis. Active practicing oncologists may find these data relevant and interesting.
My comments and suggestion.
1) The number of patients who interrupted immunotherapy because of serious complications should be clearly stated. It would be interesting to analyze the dependence of serious complications on the number of durvalumab doses used.
2) For an objective overall analysis of the data presented, it would be informative to compare the results (PFS and complications) of chemoradiotherapy + durvalumab with the results of chemoradiotherapy alone performed in the same center earlier.
3) Figure 1 is of poor quality. It needs a caption.
Round 2
Reviewer 1 Report
After the second revision of the article, I would like to congratulate the authors for their hard work in an attempt to address the shortcomings of the article. Despite the changes, the manuscript still has the same shortcomings as before. Although the minor changes have been made correctly, only one of the four major changes requested has been made. These major changes were indicated in an attempt to remedy the main flaw, which is the lack of novel or original data.
When one looks at the article again, one does not see any new data that improves or provides something different from what has already been described many times in the literature. Therefore, it is difficult to save the article and accept it for publication. No data is provided on subgroup analysis that might be different from the literature or no data is given for tumours with driver mutations for example. Therefore, I do not believe that the article in its current format is of value for publication and in my view the most correct option is "reject". Undoubtedly, the authors have made a huge effort and I think they should expand their database because the cohort of patients they have is undoubtedly very promising.
Reviewer 2 Report
I could not find substantial improvement in the Discussion section.
I think the authors should discuss more on how this study differs from the other previously reported studies.
